# Layer Collaborative Low-Rank Decomposition with Automatic Rank Search for LLM Compression

## Abstract

Large Language Models (LLMs) achieve strong performance but face deployment challenges due to high storage and memory costs. Low-rank approximation via Singular Value Decomposition (SVD) offers an effective compression solution. However, existing SVD-based methods typically compress each weight matrix independently in a layer-wise manner, ignoring the cross-layer interactions within transformer blocks and causing suboptimal performance. Moreover, conventional rank allocation strategies—either greedy or based on singular value decay—are often suboptimal, overlooking the varying sensitivity of different blocks to compression. To address these issue, we propose LC-SVD, a layer collaborative SVD framework with automatic rank search that enables adaptive low-rank compression of LLMs. Our approach includes: 1) block-wise collaborative decomposition jointly compresses all linear layers within a transformer block, preserving intra-block structural dependencies and reducing error accumulation. To improve rank allocation, we devise an error-driven rank search strategy that evaluates block sensitivity on calibration data and prioritizes capacity in more critical components via candidate configuration scoring. This ensures better accuracy under fixed resource budgets. The experimental results show that LC-SVD outperforms state-of-the-art SVD-based methods, achieving lower perplexity and higher task performance.

## 1 Introduction

Large Language Models (LLMs) have demonstrated remarkable performance across various natural language processing tasks (Brown et al., 2020; Touvron et al., 2023; Yang et al., 2025; Guo et al., 2025). However, deploying LLMs in resource-constrained environments is challenging due to the high storage and GPU memory demands from their large parameter counts. This has motivated extensive research into model compression (Ashkboos et al., 2024; Yan et al., 2025; Liu et al., 2025), with low-rank approximation methods (Chen et al., 2021; Yuan et al., 2023; Wang et al., 2025a;b; Qinsi et al., 2025) offering a promising solution. Singular Value Decomposition (SVD) (Eckart & Young, 1936) is typically used in these low-rank approximation methods to decompose weight matrices into products of small matrices, significantly reducing parameters and easing storage costs.

Naive SVD-based compression (Mao et al., 2020; Eckart & Young, 1936) approximates each weight matrix by truncating small singular values, aiming to minimize reconstruction error in isolation. However, this naive approach ignores the input activation patterns, often discarding components critical for feature transformation and leading to significant degradation of model performance. To address this limitation, recent activation-aware methods (Chen et al., 2021; Yuan et al., 2023; Wang et al., 2025a;b) incorporate activation statistics, such as covariance or scaling factors, into decomposition process. By aligning low-rank approximations with the actual data characteristics, they preserve the functional behavior of individual layers and achieve improved compression performance.

Despite these advances, existing methods suffer from two major limitations. First, they typically adopt a layer-wise compression strategy, treating each weight matrix in isolation without considering the broader network context. This local perspective overlooks the fact that LLM outputs emerge from the coordinated, nonlinear interactions of weights across layers, potentially disrupting feature compositions critical to model performance. Second, they rely on greedy layer-by-layer rank

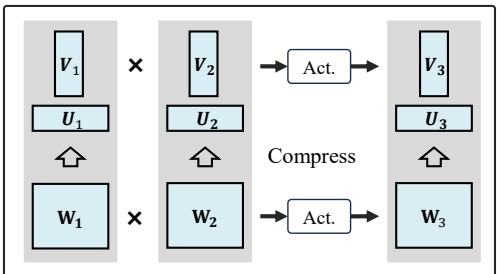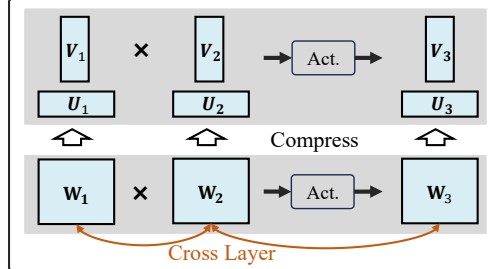

**Layer Independent SVD**    **Layer Collaborative SVD (Ours)**

Figure 1: Comparison between conventional low-rank LLM decomposition (left) and our layer collaborative low-Rank decomposition (right) within the FFN block typically adopt by LLMs (e.g. LLaMA3 and Qwen3). The conventional one compresses each weight matrix independently, while our LC-SVD integrates coordinated nonlinear interactions across FFN weights during compression.

search (Yuan et al., 2023) or pre-defined analytic formulas (Lin et al., 2025) based on heuristic statistics, to allocate rank budgets across layers. Such approaches may lead to suboptimal rank distributions. In this case, they may over-compress critical layers or under-utilize the compression potential of more robust ones, which limit the accuracy-efficiency trade-off under aggressive compression.

To address these issues, we propose a layer collaborative singular value decomposition (LC-SVD) framework that jointly optimizes block-level parameter reconstruction and data-aware rank allocation. First, to overcome the limitation of layer-wise compression, we propose block-wise collaborative decomposition, which jointly compresses the transformer block as a whole instead of multiple weight matrices in isolation. This preserves intra-block structural dependencies and reduces error accumulation across layers. Second, to enable more effective rank allocation, we propose an error-driven rank search strategy: through measuring how sensitive each block is to compression, we generate and evaluate candidate rank configurations that prioritize important blocks, leading to better performance under fixed resource budgets. The integration of these two complementary components enables LC-SVD to achieve a more accurate and adaptive low-rank compression of LLMs.

**Contributions:** 1) We identify a critical limitation in existing SVD-based LLM compression methods: their layer-wise, isolated low-rank approximation ignores the cross-layer interactions, leading to suboptimal reconstruction and significant accuracy degradation. 2) We propose LC-SVD, a layer collaborative SVD method that addresses this issue via block-wise collaborative decomposition, jointly compressing entire transformer blocks to preserve intra-block structural dependencies and mitigate error accumulation across layers. 3) We devise an error-driven rank search framework that automatically allocates rank budgets based on block-level sensitivity to compression, enabling adaptive and fine-grained rank distribution without manual tuning or predefined heuristics.

## 2 RELATED WORKS

**Large Language Model (LLMs) Compression**. LLMs face great challenges in real-world deployment due to their massive parameter count. To address this, researchers have developed model compression techniques, such as quantization, pruning, and knowledge distillation. Quantization compresses models by representing parameters in low-bit formats, with two primary approaches: 1) *quantization-aware training* (QAT) (Yao et al., 2022; Liu et al., 2024; Du et al., 2024) retrains the model after quantization to recover accuracy; and 2) *post-training quantization* (PTQ) (Frantar et al., 2023; Xu et al., 2024; Liu et al., 2025) avoids retraining by directly converting full-precision models. Pruning removes redundant parameters and is categorized into three types: 1) *unstructured pruning* (Frantar & Alistarh, 2023; Sun et al., 2024), which removes individual weights, creating irregular sparsity that requires specialized hardware; 2) *structured pruning* (Ma et al., 2023; Ashkboos et al., 2024; Sengupta et al., 2025), which removes entire components like neurons or layers, simplifying hardware implementation but often at the cost of accuracy; and 3) *semi-structured pruning* (Li et al., 2023b; Fang et al., 2024; Yan et al., 2025), which retains partial parameters based on patterns like N:M sparsity, balancing sparsity and structural regularity. Additionally, knowledge distillation

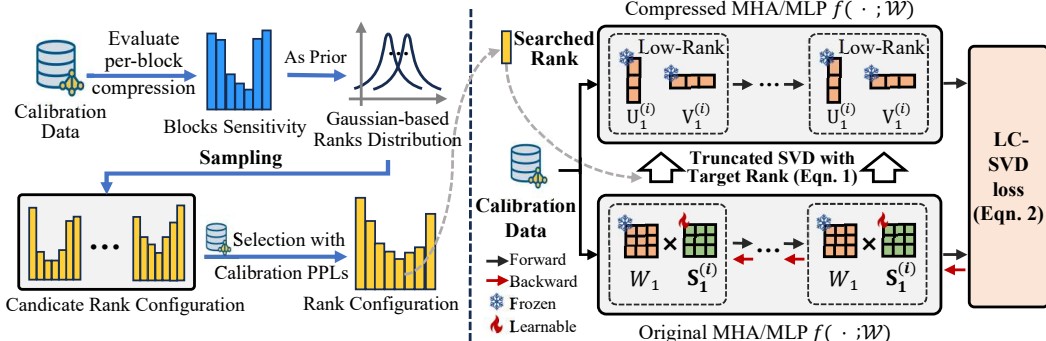

Figure 2: Illustration of LC-SVD. Our method performs layer collaborative decomposition in a data-driven pipeline. We introduce a whitening matrix $\mathbf{S}$ for each weight and iteratively optimize it by minimizing the block-level reconstruction loss (Eqn. (2)). For each block, we determine the decomposition rank via error-driven search: measuring block sensitivity by perplexity degradation, sampling candidate configurations and selecting the one with the lowest perplexity. Finally, we use optimized whitening matrix to compute the decomposed weights, resulting in the compressed block.

(KD) (Li et al., 2022; Huang et al., 2022; Gu et al., 2024) transfers knowledge from a large teacher model to a smaller student model, enabling the latter to achieve comparable performance with reduced computational costs. These techniques provide effective solutions for compressing LLMs, facilitating their efficient deployment in resource-constrained environments.

**Low-Rank Approximation for LLMs**. In addition to established methods, low-rank approximation (Hoffman, 1987) emerges as an alternative paradigm for compressing LLMs. It reduces the model size by replacing a matrix with two smaller low-rank matrices while preserving key information with Singular Value Decomposition (SVD). However, directly applying SVD (Mao et al., 2020) often degrades performance, as it overlooks the varying importance of parameters in processing inputs. To address this, activation-aware SVD methods incorporate calibration data to minimize compression-induced variations. For instance, FWSVD (Hsu et al., 2022) evaluates weight importance using Fisher information, while ASVD (Yuan et al., 2023) normalizes input impacts via diagonal scaling. Despite their utility, these methods lack theoretical optimality under rank constraints. SVD-LLM (Wang et al., 2025a;b) introduces truncation-aware whitening for per-layer optimization but achieves only local optimality. MoDeGPT (Lin et al., 2025) groups weight matrices within layers and applies tailored approximations, mitigating this issue. To address inherent information loss from truncation in SVD, Dobi-SVD (Qinsi et al., 2025) directly truncates activations by applying SVD to the activation matrix and updates weights using Incremental PCA. This method incorporates a quantization-based remapping strategy, leveraging value concentration for efficient compression. Additionally, some studies (Li et al., 2023a; Saxena et al., 2024; Zhang et al., 2024) combine low-rank approximation with other techniques, enabling higher ranks under fixed memory budgets. These advancements underscore ongoing efforts to refine low-rank methods for LLM compression.

## 3 PROPOSED METHOD

**Problem Statement**. Large Language Models (LLMs) have achieved groundbreaking performance but rely heavily on an enormous number of parameters. This results in substantial storage overhead and significant GPU memory demands during inference, posing practical bottlenecks for deployment in resource-constrained environments. The majority of these parameters (*e.g.*, exceeding 95% in LLaMA3) are in linear projection layers, with embeddings and normalization layers comprising the remainder. Let $\left\{\mathbf{W}_i \in \mathbb{R}^{d_{\text{in}} \times d_{\text{out}}}\right\}_{i=1}^{M}$ represent the weight matrices of these linear projections layers. Low-rank approximation methods (Hsu et al., 2022; Wang et al., 2025a;b; Lin et al., 2025) approximate a weight matrix $\mathbf{W}$ as the product of two low-rank matrices via singular value decomposition (SVD): $\mathbf{W} \approx \mathbf{UV}$, where $\mathbf{U} \in \mathbb{R}^{d_{\text{in}} \times r}$ and $\mathbf{V} \in \mathbb{R}^{r \times d_{\text{out}}}$, with a rank $r < \min(d_{\text{in}}, d_{\text{out}})$. Despite their effectiveness, these methods face two challenges: 1) How to leverage the input data distribution to optimize weight matrix decomposition, ensuring minimal task-performance degradation? 2) How to adaptively determine the appropriate rank $r$ for each layer $\mathbf{W}_i$?

| **Algorithm 1** Layer Collaborative Decomposition | **Algorithm 2** Error-Driven Rank Search |
|---|---|
| **Require:** LLM block $\mathcal{W}$, calibration data $\{\mathbf{X}_i\}_{i=1}^N$, the optimization budget epoch $E$ | **Require:** LLM, the target average rank $r'$, calibration data $\{\mathbf{X}_i\}_{i=1}^N$, the sampling number $T$ |
| 1: Initialize $\mathcal{S}$ via Cholesky decomposition. | 1: Measure sensitivities via per-block decompose. |
| 2: Search block rank $r_i^*$ using Algorithm 2. | 2: **for** $i \in [1, \ldots, T]$ **do** |
| 3: **for** $k \in [1, \ldots, E]$ **do** | 3:   Compute block rank expectation via Eq. (4). |
| 4:   Decompose matrix $\hat{\mathcal{W}}$ with $r_i^*$ via Eq. (1). | 4:   Sample configuration $\mathbf{r}^{(i)}$ via Eq. (5). |
| 5:   Optimize $\mathcal{S}$ with reconstruction loss in Eq. (2). | 5: **end for** |
| 6:   **if** Eq. (3) is satisfied **break** | 6: Evaluate PPL of $\{\mathbf{r}^{(i)}\}_{i=1}^T$ on calibration data. |
| 7: **end for** | 7: Select $\mathbf{r}^*$ with the best PPL metric. |
| 8: **return** $\hat{\mathcal{W}}$. | 8: **return** $\mathbf{r}^*$. |

## 3.1 METHOD OVERVIEW

In this paper, we propose a layer collaborative singular value decomposition (LC-SVD), a novel decomposition framework that jointly resolves two critical challenges in compressing large language models (LLMs). Our method consists of two key innovations: We perform reconstruction over entire transformer blocks (e.g., combined query/key/value projections) rather than individual layers. This block-level optimization preserves cross-layer dependencies and mitigates error accumulation, leading to more faithful low-rank approximation. To align decomposition with real-world data, we introduce a data-adaptive transformation (called whitening matrix) that captures the input distribution through calibration data. This matrix is dynamically updated during optimization (c.f. Section 3.2 and Algorithm 1), enabling more accurate parameter decomposition.

To adaptively determine decomposition ranks, we first compute the perplexity degradation of each block as the error under light compression, which serves as a proxy for its sensitivity. Then we sort the blocks by sensitivity, and sample a set of rank configurations that assign higher ranks to more sensitive blocks. Each candidate configuration respects the overall computational budget and follows the sensitivity ordering to prioritize important blocks. We evaluate all candidates on a small calibration dataset and select the one with the lowest perplexity as the final rank configuration (c.f. Section 3.3 and Algorithm 2). The overview of our method is shown in Figure 2.

## 3.2 LAYER COLLABORATIVE WEIGHT DECOMPOSITION

Naive SVD (Eckart & Young, 1936) directly approximates a matrix $\mathbf{W}$ by truncating its top $r$ singular values and the corresponding singular vectors, *i.e.*, $\hat{\mathbf{W}} = \text{SVD}(\mathbf{W}; r) = \mathbf{U}_{:,:r}\mathbf{\Sigma}_{:r,:r}\mathbf{V}_{:r,:}^{\top}$, where $\mathbf{U}$ and $\mathbf{V}$ are the left and right singular matrices, $\mathbf{\Sigma}$ is the diagonal matrix of singular values, and $r$ denotes the rank. However, since naive SVD does not account for the underlying distribution of the input data, it often results in suboptimal approximations and suffers from limited representational capacity. To address this, we introduce a layer collaborative SVD framework that captures the distributional characteristics of the input data through a learnable whitening matrix $\mathbf{S}$ as follows:

$$\hat{\mathbf{W}} = \text{SVD}(\mathbf{WS}; r) = \mathbf{U}_{:,:r}\mathbf{\Sigma}_{:r,:r}\mathbf{V}_{:r,:}^{\top}\mathbf{S}^{-1}, \tag{1}$$

where the weighting matrix $\mathbf{S}$ adjusts the decomposition to better reflect the intrinsic structure of the data. Let $f(\mathbf{X}_i; \mathcal{W})$ denote a Transformer block, which represents either a multilayer perceptron (MLP) or a multi-head attention (MHA) block. To obtain a promising $\mathbf{S}$, we employ gradient descent to optimize $\mathbf{S}$ by minimizing the discrepancy between the original block outputs $f(\mathbf{X}_i; \mathcal{W})$ and those generated by the decomposed weights $f(\mathbf{X}_i; \hat{\mathcal{W}})$ on a calibration dataset:

$$\min_{\mathcal{S}} \frac{1}{N} \sum_{i=1}^{N} \|f(\mathbf{X}_i; \mathcal{W}) - f(\mathbf{X}_i; \hat{\mathcal{W}})\|_F^2, \tag{2}$$

where $\mathcal{W} = \{\mathbf{W}_k\}_{k=1}^K$ and $\hat{\mathcal{W}} = \{\hat{\mathbf{W}}_k\}_{k=1}^K$ are the original weights and approximated weights of the linear layers, respectively. Each approximated weight matrix is computed via data-adaptive decomposition as $\hat{\mathbf{W}}_k = \text{SVD}(\mathbf{W}_k\mathbf{S}_k; r)$, with a learnable whitening matrix $\mathbf{S}_k$ that modulates the input space to align the low-rank approximation with the intrinsic activation distribution of the corresponding layer. The choice of initialization for $\mathbf{S}_k$ significantly affects the optimization process.

As shown in our ablation studies (see Section 4.3), different initialization strategies lead to varying convergence behaviors. We find that initializing $\mathbf{S}_k$ with a scheme that reasonably captures the activation distribution of the corresponding layer facilitates more stable and efficient optimization. Notably, we jointly optimize all whitening matrices $\mathcal{S}=\{\mathbf{S}_k\}_{k=1}^{K}$ during training.

In Eqn. (2), compared to layer-wise optimization (*i.e.*, minimizing $\|\mathbf{X}_i\mathbf{W}-\mathbf{X}_i\hat{\mathbf{W}}\|_F^2$), our layer collaborative optimization offers distinct advantages: 1) It preserves the compositional structure of non-linear transformations within the block, ensuring that the interaction between linear projections and activation functions remains intact; 2) It accounts for the cascading effect of approximation errors across multiple layers, preventing error amplification through the network and helping stabilize the whole model reconstruction quality; 3) The learnable $\mathbf{S}$ captures cross-layer statistical dependencies in the input distribution, enabling more effective whitening that benefits all constituent layers.

**Stopping Condition of Optimization**. A critical practical challenge in the optimization process lies in determining an appropriate number of iterations. Simply fixing the iteration count is inefficient, as it may either incur unnecessary computational overhead or prematurely terminate the optimization before reaching a satisfactory solution. To address this, we devise an adaptive convergence criterion $\mathcal{H}(t)$, which triggers termination of the optimization process when $\mathcal{H}(t)$ is less than a predefined threshold $\epsilon$. Specifically, $\mathcal{H}(t)$ measures the relative change between the current loss value $\mathcal{L}_t$ and the moving average loss over past $w$ iterations $(\sum_{i=t-w}^{t}\mathcal{L}_i)/w$, which filters out short-term fluctuations and focuses on the underlying convergence trends. Formally, we calculate $\mathcal{H}(t)$ as follows:

$$\mathcal{H}(t) = \left| \frac{1}{w}(\sum_{i=t-w+1}^{t}\mathcal{L}_i) - \mathcal{L}_t \right| / \mathcal{L}_0,  \tag{3}$$

where we use the initial loss $\mathcal{L}_0$ for normalization to ensure that $\mathcal{H}(t)$ remains scale-invariant across different blocks and samples, making the criterion broadly applicable.

**Stable Optimization of Whitening Matrix**. The optimization process in Eqn. (2) often encounters numerical instability during the truncated SVD gradient computation. Specifically, when two singular values $\sigma_j$ and $\sigma_i$ are close, common in high-dimensional matrices (Marčenko & Pastur, 1967), the gradient term $1/(\sigma_j^2 - \sigma_i^2)$ becomes arbitrarily large or undefined, risking premature termination and suboptimal solutions. To mitigate this issue, we adopt the $C$-th order Taylor expansion of $1/(1-r)$ to approximate $1/(\sigma_j^2 - \sigma_i^2) = (1/\sigma_j^2)/(1-r)$ as $C/\sigma_j^2$, with $r = \sigma_i^2/\sigma_j^2$ (Wang et al., 2021). Furthermore, to solve the instability caused by $1/\sigma_j^2$ when $\sigma_i \approx \sigma_j \approx 0$, we directly replace the denominator $\sigma_j^2 - \sigma_i^2$ with a small constant $\lambda$. With this simple strategy, our method effectively alleviates numerical instability and improves the robustness of optimization across different blocks.

### 3.3 Error-Driven Rank Search

Existing model compression methods typically apply a fixed rank percentage uniformly across Transformer blocks during low-rank decomposition, ignoring their varying sensitivity to compression. This often leads to over-compression of critical blocks and under-compression of redundant ones, harming performance under global efficiency constraints. To mitigate this, some approaches assign non-uniform ranks using greedy search or heuristic rules based on per-block importance scores. However, these rules rely on hand-designed functions, limiting their robustness and generalization across architectures and tasks. Instead, we propose a error-driven rank search framework that automatically discovers effective non-uniform rank configurations without relying on empirical formulas. Our approach consists of four stages: block-level error measurement, candidate generation, post-refinement of candidates, and perplexity-based selection, which we describe in detail below.

**Error Measurement of Transformer Block**. We assess the functional importance of each block by measuring its perplexity degradation as error under low-rank compression. For each block $f_i(\cdot)$, we apply SVD to its weight matrix with a fixed compression ratio (*e.g.*, 80%), while keeping all other blocks with full parameters. We then evaluate the model on a small calibration dataset and record the resulting perplexity $\text{PPL}_i$. A larger $\text{PPL}_i$ indicates higher sensitivity to compression, implying greater functional importance. Based on these values, we sort all blocks in ascending order of perplexity degradation and obtain a ranked index set $\mathcal{U} = \{u_1, u_2, \ldots, u_n\}$, where $u_1$ corresponds to the least sensitive block and $u_n$ to the most sensitive. This ranking serves as a structured prior for generating non-uniform sparsity configurations in subsequent stages.

**Candidate Rank Configurations Generation**. We seek to generate a set of candidate rank configurations $\mathcal{R} = \{\mathbf{r}^{(1)}, \ldots, \mathbf{r}^{(T)}\}$, where each $\mathbf{r}^{(t)} = (r_1^{(t)}, \ldots, r_L^{(t)})$ specifies the target rank for each block under the $t$-th configuration. These candidates should respect the importance prior, *i.e.*, assigning higher ranks to more sensitive blocks.

To encode the above order into the search process, we define a reference rank for each block based on its position in the ordered index set $\mathcal{U} = \{u_1, u_2, \ldots, u_L\}$, where $u_k$ denotes the $k$-th least sensitive block and $L$ is the total number of blocks. For block $u_k$, we set:

$$\mu_{u_k} = r'_{u_k} + B \left( \frac{k}{L} - 0.5 \right), \tag{4}$$

where $r'_{u_k}$ is the reference rank for block $u_k$, determined based on a user-specified ratio $\rho$ that reflects the overall compression preference. $B > 0$ controls the spread of rank allocation. This design ensures that blocks with higher sensitivity (larger $k$) receive larger reference ranks, while less sensitive ones are assigned smaller values, embedding the importance directly into the generation process.

However, directly adopting $\mu_{u_k}$ as the final rank would lead to deterministic allocation, limiting our ability to explore alternative configurations that may generalize better. To address this, we propose to sample around each reference rank. Specifically, for the $u_k$-th block, we draw a tentative rank from a normal distribution centered at $\mu_{u_k}$:

$$\tilde{r}_{u_k}^{(t)} \sim \mathcal{N}(\mu_{u_k}, \sigma^2), \tag{5}$$

where clipping ensures feasibility by constraining ranks within valid bounds. The variance $\sigma^2$ controls the level of perturbation, enabling exploration of non-uniform patterns beyond strict adherence to the ranking. This stochastic sampling strategy strikes a balance between exploitation (guided by importance-aware reference ranks) and exploration (via Gaussian noise), leading to a diverse yet structured candidate pool. Empirically, the resulting configurations exhibit a strong correlation between assigned ranks and block sensitivity, as measured by a high Kendall Tau coefficient.

**Post-Refinement of Rank Candidates**. After generating the candidate rank configurations, we refine each one to meet the global sparsity requirement according to the target compression ratio $\rho$. Specifically, we compute the parameter deviation $\Delta^{(t)}$ as the difference between the current parameter count of the candidate $\mathbf{r}^{(t)}$ and the desired one under $\rho$. A positive deviation indicates under-compression (too many parameters), while a negative one indicates over-compression. We then adjust the ranks to close this gap while preserving the sensitivity-aware ranking order.

- If $\Delta^{(t)} > 0$ (under-compressed), we increase compression by reducing ranks of blocks in ascending order of sensitivity (from least to most important);
- If $\Delta^{(t)} < 0$ (over-compressed), we decrease compression by restoring ranks in descending order of sensitivity, prioritizing more important blocks.

Each update adjusts a block's rank by a small step (e.g., 5%), followed by recomputation of $\Delta^{(t)}$, and terminates when the global budget is satisfied within a small tolerance.

**Final Rank Configuration Determination**. We select the final rank configuration from the set of refined candidates $\mathcal{C} = \{\mathbf{r}^{(1)}, \ldots, \mathbf{r}^{(T)}\}$ by choosing the one that achieves the lowest perplexity on a calibration dataset. This evaluation is highly efficient: computing the perplexity of each candidate takes less than one minute on a single GPU, as it involves only a forward pass without any training or fine-tuning. In our experiments, the entire selection process—evaluating all $T$ candidates—completes within one hour, making it a lightweight and practical procedure.

## 4 EXPERIMENTS

### 4.1 EXPERIMENTAL SETUP

**Models and Datasets**. We conduct our experiments on a diverse set of widely adopted large language models (LLMs), including LLaMA2-7B (Touvron et al., 2023), LLaMA3.1-8B (Grattafiori

Table 1: Comparison of SVD-based compression methods on LLaMA2-7B (Touvron et al., 2023) and LLaMA3.1-8B (Grattafiori et al., 2024) under the 80% compression ratio setting (i.e., retaining 80% of original model parameters). Perplexity (PPL) on WikiText-2 and PTB, as well as accuracy (%) on a suite of downstream tasks are reported. **Bold** and underlined entries indicate the best and second-best, respectively, across all compared compression methods.

| Methods | WikiText-2↓ | PTB↓ | ARC-c | PIQA | WinoG. | HellaS. | Com.QA | MathQA | Average↑ |
|---|---|---|---|---|---|---|---|---|---|
| LLaMA2-7B | 5.48 | 24.09 | 43.52 | 78.07 | 69.14 | 57.18 | 32.92 | 28.07 | 51.48 |
| • SVD | $1.8 \times 10^4$ | $4.2 \times 10^4$ | 22.27 | 52.23 | 48.22 | 25.97 | 20.23 | 19.43 | 31.39 |
| • ASVD | 9.51 | 165.87 | 32.34 | 71.06 | 61.17 | **46.86** | **23.10** | 24.46 | 43.16 |
| • SVD-LLM | 8.37 | 117.40 | 26.02 | 65.94 | 61.09 | 39.40 | 19.00 | 24.12 | 39.26 |
| • LC-SVD (Ours) | **6.95** | **53.13** | **33.96** | **71.22** | **64.48** | 44.74 | 20.39 | **25.33** | **43.35** |
| LLaMA3.1-8B | 7.22 | 11.45 | 51.79 | 80.09 | 74.27 | 59.05 | 77.15 | 39.53 | 63.65 |
| • SVD | $2.9 \times 10^5$ | $2.5 \times 10^5$ | 23.21 | 52.72 | 50.43 | 25.62 | 20.39 | 18.53 | 31.82 |
| • ASVD | 2244.04 | 4719.57 | 20.56 | 55.88 | 48.86 | 26.67 | 19.25 | 19.40 | 31.77 |
| • SVD-LLM | 16.65 | 148.41 | 26.96 | 62.19 | 59.98 | 38.01 | 26.86 | 25.63 | 39.94 |
| • LC-SVD (Ours) | **12.37** | **117.40** | **32.42** | **68.12** | **63.77** | **41.28** | **47.01** | **28.17** | **46.79** |

et al., 2024), and a larger-scale model Qwen3-32B (Yang et al., 2025) to evaluate the scalability of our method. In terms of datasets, we follow standard practices in SVD-based methods (Qinsi et al., 2025; Wang et al., 2025b) for calibration and evaluation. For the calibration dataset, we use 256 training samples from Wikitext-2, each containing 2048 tokens. For evaluating the compressed models, we assess performance on two language modeling datasets: Wikitext-2 (Merity et al., 2017) and PTB (Marcus & Marcinkiewicz, 1993). Additionally, we employ the LM-Evaluation-Harness framework (Gao et al., 2021) to evaluate all methods on six classification tasks, covering common-sense understanding, science question answering, and mathematical problem solving. Further details regarding these datasets are provided in Section B.1 in the supplementary.

**Baselines**. We conduct comprehensive comparisons between our proposed LC-SVD and several state-of-the-art SVD-based LLM compression methods, including Naive SVD, which directly truncates singular values of weight matrix; SVD LLM (Wang et al., 2025b) and ASVD (Yuan et al., 2023), both activation-aware variants that incorporate activation statistics of weight during decomposition. All baselines are implemented using their official implementations with default hyperparameters. To ensure a fair comparison, we use the same calibration dataset, apply identical evaluation protocols, and do not consider additional recovery fine-tuning for all methods.

**Implementation Details**. All experiments are conducted on four NVIDIA A800 GPUs with PyTorch. In the layer collaborative SVD optimization, we employ an AdamW optimizer with a learning rate of 0.01, and each block is optimized for up to 50 epochs. The batch size is set to the size of entire calibration dataset (256 samples), meaning one update per epoch. We adopt an adaptive stopping criterion based on the relative change in loss over the past 5 iterations, terminating when this change falls below $\epsilon = 1 \times 10^{-5}$. For the error-driven rank search, we sample candidate rank configurations from a Gaussian distribution centered at block-specific reference ranks, with a variance of 0.03. Additional implementation details are provided in Section B.2 in the supplementary.

## 4.2 COMPARISONS WITH STATE-OF-THE-ART METHODS

**Comparisons with SVD-Based Compression Methods**. We compare LC-SVD with state-of-the-art SVD-based methods, including Naive SVD, ASVD (Yuan et al., 2023), and SVD-LLM (Wang et al., 2025b), on LLaMA2-7B (Touvron et al., 2023), LLaMA3.1-8B (Grattafiori et al., 2024), under the 80% compression ratio setting (*i.e.*, the compressed model retains 80% of the original number of parameters). Due to space limitations, additional results under the 60% and 40% compression ratios are presented in Section C.1 of the supplementary material.

In Table 1, our method consistently achieves the lowest perplexity on both WikiText-2 and PTB across most models and the highest average accuracy on downstream tasks across all models. For LLaMA2-7B, the perplexity of our LC-SVD is lower than that of SVD-LLM(6.95 *vs.* 8.37 on WikiText-2 and 53.13 *vs.* 117.40 on PTB) and ASVD (6.95 *vs.* 9.51 on WikiText-2 and 53.13 *vs.* 165.87 on PTB). In terms of downstream task performance, LC-SVD achieves an average accuracy

Table 2: Ablation study on the initialization for the whitening matrix in the layer collaborative SVD framework. We compare random initialization with Cholesky decomposition-based initialization on LLaMA2-7B under 80% compression ratio. Ours* denotes the ablation setting where only the first component is applied; the random initialization variant uses the same framework but replaces the initialization with a random one.

| Methods | WikiText-2↓ | PTB↓ | ARC-c | PIQA | WinoG. | HellaS. | Com.QA | MathQA | Average↑ |
|---|---|---|---|---|---|---|---|---|---|
| LLaMA2-7B | 5.48 | 24.09 | 43.52 | 78.07 | 69.14 | 57.18 | 32.92 | 28.07 | 51.48 |
| • Random Initilization | 10.94 | 96.81 | 27.73 | 66.65 | 61.96 | 39.96 | 19.57 | 23.99 | 39.98 |
| • LC-SVD (Ours*) | **7.09** | **53.17** | **32.98** | **70.44** | **63.59** | **43.13** | **19.90** | **26.06** | **42.68** |

Table 3: Comparison of SVD-based compression methods on the large-scale instruction-tuned Qwen3-32B (Yang et al., 2025) under the same 80% compression ratio setting and downstream evaluation protocol as in Table 1. **Bold** values indicate the best among all compression methods. "Wiki.-2" denotes the WikiText-2 dataset.

| Methods | Wiki.-2↓ | PTB↓ | WinoG.↑ | HellaS.↑ |
|---|---|---|---|---|
| Qwen3-32B | 7.62 | 11.81 | 73.24 | 63.87 |
| • SVD-LLM | 10.10 | 43.87 | 70.24 | 54.43 |
| • LC-SVD (Ours) | **8.91** | **26.61** | **72.45** | **56.13** |

Table 4: Ablation study on the proposed components of our LC-SVD on LLaMA2-7B under 80% compression ratio. We report Perplexity on WikiText-2. Note that the baseline uses default whitening matrix initialization and weight decomposition without any proposed components.

| Method | PPL↓ |
|---|---|
| Baseline | 8.37 |
| +Error-Driven Rank Search (Sec. 3.3) | 8.26 |
| +Layer collaborative SVD (Sec. 3.2) | 7.09 |
| LC-SVD (Ours) | **6.95** |

of 43.35, surpassing ASVD (43.16) and SVD-LLM (39.26). For LLaMA3.1-8B, LC-SVD further widens the performance gap, achieving a perplexity of 12.37 on WikiText-2 and 117.40 on PTB, compared to 16.65 and 148.41 for SVD-LLM and 2244.04 and 4719.57 for ASVD. The average accuracy on downstream tasks reaches 46.79, significantly higher than that of SVD-LLM (39.94) and ASVD (31.77). These performance improvements primarily stem from our method's ability to incorporate block-level activation information into the SVD process and leverage the heterogeneity in compression sensitivity across blocks. Consequently, our method leads to better alignment between the compressed and original models in terms of both representation fidelity and task performance.

**Comparisons of SVD-Based Compression Methods on Larger Scale LLM Qwen3-32B**. To evaluate the scalability of LC-SVD, we further conduct experiments on a much larger model, Qwen3-32B, under the same 80% compression ratio. In Table 3, our method achieves a perplexity of 8.91 on WikiText-2 and 26.61 on PTB, significantly outperforming the strong SVD-based baseline SVD-LLM, which obtains 10.10 and 43.87 on the two datasets, respectively. Due to computational constraints, we evaluate on WinoGrande and HellaSwag downstream tasks, which assess commonsense and semantic reasoning. While LC-SVD shows a slight drop compared to the original model, it outperforms SVD-LLM by +2.21 and +1.70 points, respectively. This consistent performance gain demonstrates that LC-SVD remains effective even when applied to models with over 30 billion parameters. These results indicate that the benefits of leveraging block-level data distribution and heterogeneous sensitivity are not limited to smaller models, but generalize well to large-scale one.

## 4.3 ABLATION STUDY

**Ablation on different initializations of whitening matrix**. We ablate the whitening matrix initialization in our layer-collaborative SVD framework, comparing random initialization with our default one based on the Cholesky decomposition of the input covariance, which captures the input data statistics. To isolate the effect of initialization, we disable error-driven rank search and apply a uniform compression ratio 80% and ensure both methods are trained under the same optimization budget (i.e., identical number of update steps). As shown in Table 2, random initialization yields worse perplexity on WikiText-2 (10.94) than SVD-LLM (8.37) and only marginal downstream gains (39.98 average task accuracy). In contrast, the default initialization achieves significantly better results (7.09 / 53.17 PPL, 42.68 average accuracy), confirming that a data-aware initialization provides a better starting point for effective optimization of weight decomposition.

Table 5: Effect of stopping condition in layer collaborative SVD on LLaMA2-7B at 80% compression.

| Stop Cond. | WT-2 PPL↓ | Avg. Epochs↓ |
|:---:|:---:|:---:|
| × | 7.09 | 50.0 |
| ✓ | 7.09 | 39.9 |

Table 6: Effect of different learning rates in layer collaborative SVD on LLaMA2-7B at 80% compression.

| Lr | 0.1 | 0.01 | 0.001 | 0.0001 |
|:---|:---:|:---:|:---:|:---:|
| Wiki.-2 PPL | 7.25 | **7.09** | 7.18 | 7.42 |

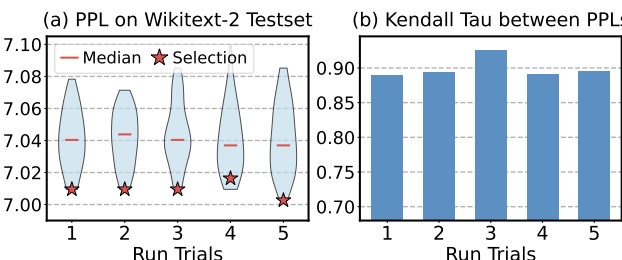

Figure 3: (a) Distribution of PPL on the calibration dataset across 5 trials (each yields 40 rank configurations). (b) Kendall Tau correlation between the PPL on the calibration dataset and the Wikitext-2 testset for each trial, demonstrating the calibration-to-test consistency of configurations.

**Ablation on different components of LC-SVD**. We conduct an ablation study on LLaMA2-7B under the 80% compression ratio, using WikiText-2 perplexity to assess the contribution of each core component in LC-SVD. The baseline model only uses the default initialization of the whitening matrices. As shown in Table 4, adding the layer collaborative SVD module reduces the perplexity from 8.37 to 7.09, demonstrating its effectiveness in aligning the decomposition with the input data distribution. Adding only the error-driven rank search also improves performance (8.26 *vs.* 8.37), indicating its role in better rank allocation. LC-SVD combines both components and achieves the best performance (6.95 *vs.* 8.37), outperforming either ablated variant. This result confirms that both components are essential and complementary: whitening optimization improves local reconstruction fidelity, while sensitivity-aware rank search ensures global compression efficiency.

**Ablation on Stopping Conditions**. We investigate the impact of the adaptive convergence criterion in the layer collaborative SVD component, disabling all other components to isolate its impact. As shown in Table 5, enabling the stopping condition reduces the average number of optimization epochs from 50.0 to 39.9, while achieving the same perplexity (7.09). These results demonstrate that the criterion safely terminates optimization early without compromising reconstruction quality, improving the optimization efficiency with no performance cost.

**Ablation on Randomness in Rank Search**. We evaluate rank configurations sampled with different random seeds to assess the robustness of our Sensitivity-Aware Rank Search framework to sampling variability. As shown in Figure 3, while worst-case candidates approach uniform compression performance (7.09), the median and best-performing results show low variance across different seeds, demonstrating the stability of our search framework. Furthermore, the high Kendall $\tau$ correlation (0.85–0.95) between the perplexity on calibration dataset and the perplexity on Wikitext-2 dataset confirms the reliability of the calibration metric. This strong correlation ensures consistent selection of near-optimal configurations (marked with stars) for Wikitext-2. In addition, these results validate the effectiveness of using calibration data to guide rank selection.

**Ablation on Learning Rate**. We analyze the sensitivity of the layer collaborative SVD component in isolation to different learning rates, including 0.1, 0.01, 0.001, and 0.0001. As shown in Table 6, the performance peaks at 0.01, achieving a perplexity of 7.09, while higher or lower learning rates lead to degraded results (7.25 and 7.42), confirming the importance of appropriate learning rate selection. Notably, we use 0.01 uniformly across all models without per-model tuning.

## 5 CONCLUSION

In this paper, we propose LC-SVD, a layer collaborative SVD with error-driven rank search for large language model compression. Our method introduces a learnable blockwise SVD to incorporate input data distribution into decomposition, and a sensitivity-aware rank search to adaptively allocate ranks across layers. Experiments show that LC-SVD consistently outperforms state-of-the-art compression methods while preserving maintaining model performance. Our work highlights the importance of data and sensitivity awareness in low-rank approximation for LLM deployment.

## REPRODUCIBILITY STATEMENT

In this work, we implement our LC-SVD on different LLMs with WikiText-2 calibration dataset. Reproducing all results in our method depends on the following three aspects:

1. **DATASET.** The datasets employed in all method implementations and experimental evaluations are introduced in the first paragraph of Section 4.1 and Section B.1 in the supplementary.

2. **MODELS.** All LLMs adopted for compression are publicly available on Hugging Face using the provided model names in the second paragraph of Section 4.1.

3. The implementation details of LC-SVD and the compared methods are described in the third paragraph of Section 4.1, with additional protocol provided in Section B.2 in the supplementary.

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

# SUPPLEMENTARY MATERIALS

In this supplementary, we provide further details on our method LC-SVD and additional experimental results. We organize the content as follows.

- In Section A, we describe the usage of large language models in the writing process of this paper.
- In Section B, we provide more details on the implementations of evaluation and all methods.
- In Section C, we show more experimental results further validating the effectiveness of LC-SVD.
- In Section D, we present details about the stable optimization in layer collaborative decomposition.

## A  LLM USAGE STATEMENT

We use LLM models solely as a tool to improve the clarity and fluency of the writing. All intellectual contributions, including research ideation and technical development, remain entirely our own. The LLMs did not contribute to the scientific content or conceptual direction of this work.

## B  MORE IMPLEMENTATION DETAILS

### B.1  MORE DETAILS ON DATASET AND EVALUATION METRICS

**ARC-Challenge** (Clark et al., 2018) is a subset of the AI2 Reasoning Challenge (ARC) dataset, designed to test fine-grained scientific reasoning in language models. It consists of 2,590 multiple-choice questions from grade-school science exams, selected based on their difficulty (i.e., those not easily answered by retrieval-based methods). The dataset emphasizes deep reasoning over simple fact retrieval, requiring models to integrate background knowledge and infer causal relationships.

**PIQA** (Bisk et al., 2020) (Physical Interaction: Question Answering) focuses on physical commonsense reasoning, where models must predict the most plausible solution to everyday physical tasks. The dataset contains over 16,000 question-answer pairs, each with two candidate answers—one correct and one subtly incorrect. PIQA challenges models to understand implicit physical knowledge (e.g., "What is the best way to open a tight jar?"), which is rarely stated explicitly in text.

**Winogrande** (Sakaguchi et al., 2020) is a large-scale dataset designed to evaluate commonsense reasoning through pronoun resolution. Inspired by the Winograd Schema Challenge, it contains 44,000 multiple-choice questions, each requiring the model to identify the correct referent of an ambiguous pronoun using contextual and world knowledge. For instance, in the sentence "The trophy doesn't fit in the suitcase because it is too small," the model must determine whether "it" refers to the trophy or the suitcase. Unlike earlier Winograd-style benchmarks, Winogrande is constructed with adversarial filtering to reduce annotation artifacts and statistical shortcuts. Distractors are chosen to be grammatically valid and contextually plausible, ensuring that the task cannot be solved through surface patterns alone. This design emphasizes genuine reasoning over spurious correlations, making the benchmark resistant to overfitting by large pretrained models.

**HellaSwag** (Zellers et al., 2019) is a large-scale benchmark that tests a model's ability to infer plausible continuations of everyday situations. It contains 39,905 multiple-choice questions, where each instance presents an incomplete narrative or activity description with four candidate endings. Among them, one is the correct continuation, while the remaining three are adversarially generated by strong language models to appear coherent and contextually relevant. The dataset draws content from two domains: video activity descriptions and grounded narratives. This design ensures coverage of both physical commonsense (e.g., human actions and object interactions) and abstract event reasoning in natural text. Unlike simpler cloze-style benchmarks, HellaSwag emphasizes subtle distinctions between likely and unlikely outcomes, requiring models to rely on nuanced commonsense knowledge rather than superficial cues. Due to its adversarial construction and focus on realistic scenarios, HellaSwag has become a challenging testbed for evaluating pretrained language models. It highlights limitations of systems that depend on shallow heuristics, pushing research toward better modeling of fine-grained commonsense and event plausibility.

**CommonsenseQA** (Talmor et al., 2019) is a benchmark designed to test commonsense reasoning beyond factual recall. It contains 12,102 multiple-choice questions, each with five candidate answers, requiring models to select the most plausible choice based on everyday knowledge such as causes, properties, or typical uses. The dataset is built from ConceptNet (Speer et al., 2017). Question templates are created from ConceptNet triples (e.g., `Causes`(*exercise*, *sweating*)), and annotators transform them into natural questions. Distractor options are intentionally chosen to be plausible yet incorrect, preventing simple reliance on lexical overlap or frequency cues. Questions are categorized into five semantic types, ensuring diverse coverage of commonsense relations. Unlike standard reading comprehension datasets, CommonsenseQA requires reasoning across indirect connections, often demanding integration of relational knowledge that is not explicitly stated.

**MathQA** (Amini et al., 2019) is a math word problem-solving dataset that evaluates a model's ability to perform multi-step arithmetic reasoning in natural language. It contains over 23,000 problems from standardized exams, each paired with a multiple-choice format and an executable Python program as the rationale. MathQA requires not only numerical understanding but also the ability to map linguistic descriptions to formal operations.

**Perplexity (PPL)** is a standard metric for evaluating language modeling performance, defined as the exponentiated average negative log-likelihood of the test sequence. Lower perplexity indicates better fluency and generalization in modeling the data distribution. We report PPL on the test splits of two widely used benchmark datasets: **WikiText-2** (Merity et al., 2017) and **Penn Treebank (PTB)** (Marcus & Marcinkiewicz, 1993). WikiText-2 is a large-scale corpus derived from Wikipedia articles, characterized by long-range dependencies and rich syntactic diversity. PTB is a well-established dataset composed of Wall Street Journal text, known for its clean annotation and balanced vocabulary distribution. We evaluate perplexity on both datasets to assess how well the compressed models preserve the language modeling capability acquired during pre-training, following standard practice in model compression for LLMs.

### B.2 MORE EXPERIMENTAL PROTOCOLS

**Layer Collaborative Weight Decomposition Component.** We optimize each block using AdamW with a learning rate of $0.01$ for up to 50 epochs, with a batch size equal to the full calibration dataset (one update per epoch). Convergence is determined by an adaptive stopping criterion: optimization halts when the relative loss change over the past five iterations falls below $\epsilon = 1 \times 10^{-5}$. To ensure numerical stability during truncated SVD, we approximate unstable gradient terms using a 10-th order Taylor expansion and introduce a small constant $\lambda = 1 \times 10^{-10}$ to prevent division by near-zero singular values. All SVD operations are initially performed in 32-bit precision to ensure numerical stability; afterward, the decomposed low-rank matrices are converted back to the original model's precision to maintain compatibility with downstream inference and evaluation.

**Error-Driven Rank Search Component.** We sample candidate rank configurations from Gaussian distributions centered at block-specific reference ranks, with variance $0.03$. To promote diversity in the sampled configurations, we we vary the scaling parameter $B \in \{0.1, 0.2, \ldots, B_{\max}\}$ in increments of $0.1$. For each $B$, we compute reference ranks and sample 10 configurations. $B_{\max}$ is set such that either: (1) the most sensitive block retains a rank yielding near-original parameter count, or (2) the least sensitive block drops below 5% of its original parameters. This ensures a diverse yet meaningful exploration of rank allocations, balancing representational fidelity and compression efficiency across blocks.

**Baseline and Evaluation.** All baseline methods are implemented using their official codebases, with default hyperparameters and the same calibration dataset to ensure a fair comparison. All downstream evaluations are performed using the LM-Evaluation-Harness framework (Gao et al., 2021), under zero-shot setting and using the default provided hyperparameters.

## C MORE EXPERIMENTAL RESULTS

### C.1 MORE COMPARISONS OF SVD-BASED METHODS

We present additional results under more aggressive compression ratios to further evaluate the effectiveness of LC-SVD. We report performance on LLaMA2-7B (Touvron et al., 2023), LLaMA3.1-

Table A: Comparison of SVD-based compression methods on LLaMA2-7B (Touvron et al., 2023), LLaMA3.1-8B (Grattafiori et al., 2024) and Qwen3-8B (Yang et al., 2025) at 60% compression ratio (i.e., retaining 60% of original parameters). Perplexity (PPL) on WikiText-2 and PTB (lower is better), and accuracy (%) on downstream tasks (higher is better) are reported. **Bold** and underlined denote best and second-best results. N/A indicates values that are unavailable due to numerical instability leading to NaN during evaluation. Notice that Qwen3-8B is an instruction-tuned LLM.

| Methods | WikiText-2↓ | PTB↓ | ARC-c | PIQA | WinoG. | HellaS. | Com.QA | MathQA | Average↑ |
|---|---|---|---|---|---|---|---|---|---|
| LLaMA2-7B | 5.48 | 24.09 | 43.52 | 78.07 | 69.14 | 57.18 | 32.92 | 28.07 | 51.48 |
| • SVD | $6.1 \times 10^4$ | $5.6 \times 10^4$ | 20.99 | 52.94 | 49.88 | 25.65 | 20.15 | 18.19 | 31.30 |
| • ASVD | N/A | N/A | 21.59 | 51.74 | 50.51 | 25.92 | 19.66 | 20.20 | 31.60 |
| • SVD-LLM | 16.14 | 525.91 | 21.42 | 58.38 | 55.64 | 30.71 | **20.16** | 22.14 | 34.74 |
| • LC-SVD (Ours) | **9.92** | 111.93 | **25.68** | **63.71** | **59.67** | **36.13** | 19.41 | **23.48** | **38.01** |
| LLaMA3.1-8B | 7.22 | 11.45 | 51.79 | 80.09 | 74.27 | 59.05 | 77.15 | 39.53 | 63.65 |
| • SVD | $1.3 \times 10^5$ | $1.1 \times 10^5$ | 21.03 | 52.29 | 49.41 | 25.52 | 19.90 | 17.52 | 30.95 |
| • ASVD | $1.7 \times 10^4$ | $1.6 \times 10^4$ | 21.50 | 52.01 | 49.80 | 25.91 | 19.00 | 18.22 | 31.07 |
| • SVD-LLM | 182.47 | 7003.51 | 19.97 | 53.97 | 51.30 | 27.05 | 19.00 | 20.80 | 32.02 |
| • LC-SVD (Ours) | **28.38** | 402.15 | **21.59** | **60.07** | **56.51** | **31.21** | 20.15 | **22.85** | **35.40** |
| Qwen3-8B | 9.78 | 15.44 | 55.80 | 76.88 | 67.88 | 57.15 | 78.71 | 49.61 | 64.34 |
| • SVD | $4.5 \times 10^7$ | $3.5 \times 10^7$ | 21.93 | 52.39 | 49.41 | 25.55 | **20.07** | 17.59 | 31.16 |
| • ASVD | 1777.87 | 4605.74 | 21.25 | 54.79 | 50.67 | 26.37 | 19.08 | 21.01 | 32.19 |
| • SVD-LLM | 29.53 | 287.64 | 21.33 | 59.63 | 53.99 | 31.47 | 19.57 | 22.38 | 34.73 |
| • LC-SVD (Ours) | **21.47** | 178.38 | **25.34** | **61.64** | **59.91** | **34.78** | 19.66 | **24.12** | **37.57** |

8B (Grattafiori et al., 2024), and Qwen3-8B (Yang et al., 2025) at 60% and 40% parameter retention ratios, respectively. All evaluation metrics include perplexity on WikiText-2 and PTB and accuracy on six downstream tasks, with the average accuracy reported in the final column.

**Results at 60% Compression Ratio (Table A).** We evaluate all methods under a 60% parameter retention setting. As shown in Table A, LC-SVD consistently outperforms all baseline methods across all models in both language modeling and downstream task performance. For instance, on LLaMA2-7B, LC-SVD achieves a perplexity of 9.92 on WikiText-2 and 111.93 on PTB, significantly better than SVD-LLM (16.14 and 525.91), while attaining the highest average accuracy of 38.01, surpassing the second-best method by 3.2 points. Notably, ASVD fails to produce valid results (N/A) due to numerical instability under this compression level, highlighting its sensitivity to aggressive rank reduction. The strong performance of LC-SVD under moderate compression confirms that our proposed components effectively preserve the model functionality.

**Results at 40% Compression Ratio (Table B).** Under the more aggressive 40% parameter retention setting, the performance gap across methods narrows on downstream tasks, yet LC-SVD still achieves the best overall results across all models. This narrowing is expected and can be attributed to the severe degradation in language modeling capability common to all SVD-based methods. As reflected by the sharp increase in perplexity, heavily compressed models struggle to capture the underlying data distribution, resulting in outputs that provide little meaningful signal for downstream classification—performances often approach those of random guessing. In contrast, LC-SVD better preserves critical functional mappings, thereby retaining more of the original model's language modeling capacity and achieving performance that slightly surpasses random guessing.

Notably, on Qwen3-8B, our method exhibits a slightly higher perplexity on WikiText-2 compared to SVD-LLM. We hypothesize that this stems from the nature of Qwen3-8B as an instruction-tuned model with strong reasoning ("thinking") capabilities. Such models are optimized for structured, task-oriented generation, which diverges significantly from the plain, factual text in WikiText-2. Consequently, the input activation patterns derived from this domain may not accurately reflect the model's intended operational behavior. Aggressive alignment to such a mismatched calibration set can inadvertently distort the internal representations, leading to performance degradation on the original modeling objective. This observation highlights a potential limitation of calibration-based compression methods when applied to instruction-tuned LLMs on non-representative data, and underscores the importance of domain alignment in the calibration process.

Table B: Comparison of SVD-based compression methods on LLaMA2-7B (Touvron et al., 2023), LLaMA3.1-8B (Grattafiori et al., 2024) and Qwen3-8B (Yang et al., 2025) at 40% compression ratio (i.e., retaining 40% of original parameters). Perplexity (PPL) on WikiText-2 and PTB (lower is better), and accuracy (%) on downstream tasks (higher is better) are reported. **Bold** and underlined denote best and second-best results. N/A indicates values that are unavailable due to numerical instability leading to NaN during evaluation. Notice that Qwen3-8B is an instruction-tuned LLM.

| Methods | WikiText-2↓ | PTB↓ | ARC-c | PIQA | WinoG. | HellaS. | Com.QA | MathQA | Average |
|---|---|---|---|---|---|---|---|---|---|
| LLaMA2-7B | 5.48 | 24.09 | 43.52 | 78.07 | 69.14 | 57.18 | 32.92 | 28.07 | 51.48 |
| • SVD | $6.5 \times 10^4$ | $7.7 \times 10^4$ | **23.21** | 53.32 | 51.54 | 25.39 | 19.33 | 19.03 | 31.97 |
| • ASVD | N/A | 12571.71 | 23.12 | 51.74 | 51.62 | 25.88 | **19.82** | 19.03 | 31.87 |
| • SVD-LLM | 89.92 | 1625.10 | 21.93 | 53.32 | 50.12 | 26.83 | 19.57 | 20.44 | 32.03 |
| • LC-SVD (Ours) | **26.22** | **331.4** | 20.73 | **56.20** | 52.49 | 28.83 | 19.25 | **22.65** | **33.36** |
| LLaMA3.1-8B | 7.22 | 11.45 | 51.79 | 80.09 | 74.27 | 59.05 | 77.15 | 39.53 | 63.65 |
| • SVD | $2.4 \times 10^5$ | $2.5 \times 10^5$ | **21.25** | 53.48 | 49.88 | 25.61 | 19.16 | 19.30 | 31.45 |
| • ASVD | $2.1 \times 10^4$ | $1.6 \times 10^4$ | 21.16 | 52.12 | 48.70 | 25.73 | **19.66** | 18.29 | 30.94 |
| • SVD-LLM | $2.6 \times 10^4$ | $7.6 \times 10^4$ | 19.37 | 53.43 | 51.14 | 25.79 | 19.57 | 18.53 | 31.30 |
| • LC-SVD (Ours) | **145.00** | **1622.70** | 18.94 | 53.54 | 51.93 | 26.67 | 19.33 | **21.17** | 31.93 |
| Qwen3-8B | 9.78 | 15.44 | 55.80 | 76.88 | 67.88 | 57.15 | 78.71 | 49.61 | 64.34 |
| • SVD | $4.0 \times 10^7$ | $9.6 \times 10^7$ | **24.40** | 51.36 | 50.99 | 25.69 | 18.76 | 19.50 | 31.78 |
| • ASVD | $3.3 \times 10^5$ | $4.1 \times 10^5$ | 20.39 | 52.99 | 50.28 | 25.63 | **21.46** | 20.74 | 31.92 |
| • SVD-LLM | **177.48** | 4596.38 | 18.86 | 54.41 | 51.07 | 27.15 | 19.49 | 21.27 | 32.04 |
| • LC-SVD (Ours) | 181.39 | **1395.22** | 18.00 | **55.28** | 53.28 | 28.23 | 19.57 | 21.54 | **32.65** |

Table C: Comparisons of pruning methods on LLaMA2-7B (Touvron et al., 2023) under 80% compression ratio. All methods are evaluated without additional recovery fine-tuning. **Bold** and underlined values indicate the best and second-best.

| Methods | WikiText-2↓ | PTB↓ | ARC-c | PIQA | WinoG. | HellaS. | Com.QA | MathQA | Average↑ |
|---|---|---|---|---|---|---|---|---|---|
| LLaMA2-7B | 5.48 | 24.09 | 43.52 | 78.07 | 69.14 | 57.18 | 32.92 | 28.07 | 51.48 |
| • LLM Pruner | 8.46 | 54.59 | 32.20 | **73.19** | 59.51 | 43.97 | 19.90 | 21.88 | 41.78 |
| • SliceGPT | **6.84** | 53.96 | 24.32 | 70.07 | 64.47 | 25.71 | 19.57 | **25.79** | 38.32 |
| • LC-SVD (Ours) | 6.95 | **53.13** | 33.96 | 71.22 | 64.48 | 44.74 | 20.39 | 25.33 | **43.35** |

## C.2 COMPARISONS OF PRUNE-BASED COMPRESSION METHODS

We compare LC-SVD with state-of-the-art pruning methods SliceGPT (Ashkboos et al., 2024) and LLM Pruner (Ma et al., 2023) on LLaMA2-7B under 80% compression without fine-tuning. As shown in Table C, LC-SVD achieves a perplexity of 6.95 on WikiText-2 and 53.13 on PTB, outperforming LLM Pruner and SliceGPT in PTB while remaining competitive on WikiText-2. For downstream tasks, LC-SVD achieves an average accuracy of 43.35%, on par with LLM Pruner (41.77%) and notably higher than SliceGPT (38.32%).

## C.3 COMPARISONS OF GENERATION OUTPUTS BETWEEN ORIGINAL AND COMPRESSED LLMs

To assess the impact of compression on high-level reasoning and language fluency, we compare the output behavior of the original Qwen3-32B and our compressed model under 80% compression ratio. As shown in Table D, both models generate well-structured, step-by-step reasoning with fluent and grammatically correct language. The compressed model successfully preserves the reasoning behavior, accurately identifying key problem elements and arriving at correct final answers. This indicates that our method maintains model's ability to perform complex, multi-step inference.

Table D: Comparison of outputs between the original Qwen3-32B and our compressed model (80% parameter retention). Both models generate coherent, step-by-step reasoning traces. The compressed model maintains fluent language generation and logical consistency, demonstrating that LC-SVD preserves high-level reasoning capabilities despite parameter reduction.

| | Qwen3-32B (Origin Model) | Qwen3-32B (Compressed by our LC-SVD) |
|---|---|---|
| **Question** | What might a successful writing program cause? A. bugs B. frustration C. need to integrate D. loop E. satisfaction | |
| **Answer** | `<think>` Okay, let's see. The question is asking what a successful writing program might cause. The options are bugs, frustration, need to integrate, loop, satisfaction. Hmm. First, I need to understand what a writing program is. A writing program could refer to a software tool that helps with writing, like a word processor or a more specialized tool for writers. Or maybe it's a program in the sense of a plan or curriculum for improving writing skills. The question is a bit ambiguous, but I'll have to go with the most likely interpretation. Assuming it's a software tool, a successful one would probably make writing easier, more efficient, or more enjoyable. So the possible outcomes would be positive. Let's look at the options. Bugs are usually negative, so a successful program might not cause bugs. Frustration is also negative, so if the program is successful, it should reduce frustration, not cause it. Need to integrate—maybe if the program is part of a larger system, but I'm not sure. Loop is a programming term, but in the context of a writing program causing a loop. `··· <think>` A successful writing program, whether a software tool or a structured curriculum, is designed to enhance the writing process. Its success is typically measured by positive outcomes such as improved efficiency, usability, or user experience. Among the given choices: Bugs and frustration are negative outcomes that would indicate failure, not success. Loop refers to a programming concept and is unrelated to the broader impact of a writing program. Need to integrate might describe a technical requirement but does not directly reflect the result of success. Satisfaction is the most logical outcome, as a successful program would likely fulfill user needs, streamline tasks, and create a positive experience. **Answer**: satisfaction. | `<think>` Okay, so I need to figure out what a successful writing program might cause. The options are bugs, frustration, need to integrate, loop, and satisfaction. Hmm, I need to think about what each of these words mean and how they relate to writing programs. Bugs are things that cause problems in software. Maybe if the program has bugs, it's not successful, but maybe a successful program might still have bugs. But maybe the question is asking what a successful writing program causes. So maybe it's something positive, like satisfaction. Frustration is negative, so maybe it's not something a successful writing program would cause. Need to integrate might mean that the program needs to integrate with other things, maybe other programs or systems. Loop might mean the program loops back on itself, like a loop in code, but maybe it's something else. Satisfaction is positive, so maybe that's the answer. A successful program might cause satisfaction. But maybe the other options are also possible. `··· <think>` The best answer is E satisfaction because a successful writing program would likely cause the writer to feel satisfaction in their work. satisfaction is positive and fits the idea of success . The other options are not as positive , and maybe bugs or frustration are not things a successful program would cause . So E satisfaction is the best answer. |

| | Qwen3-32B (Origin Model) | Qwen3-32B (Compressed by our LC-SVD) |
|---|---|---|
| **Question** | James decides to run 3 sprints 3 times a week. He runs 60 meters each sprint. How many total meters does he run a week? | |
| **Answer** | `<think>` Okay, let me try to figure out how many meters James runs in a week. So, the problem says he does 3 sprints, 3 times a week, and each sprint is 60 meters. Hmm, let me break this down step by step.

First, let's see how much he runs in one session. If he does 3 sprints and each sprint is 60 meters, then for one session it would be 3 multiplied by 60. Let me write that out: 3 * 60. Let me calculate that... 3 times 60 is 180 meters per session. Okay, that makes sense.

Now, he does this 3 times a week. So, if each session is 180 meters, then in a week he does 3 sessions. So, I need to multiply the meters per session by the number of sessions per week. That would be 180 * 3. Let me do that math.

· · ·

I think that's all there is to it. The key was to figure out the total number of sprints he does in a week and then multiply by the distance per sprint. Alternatively, calculate per session and then multiply by the number of sessions. Either way, the answer is 540 meters. I don't see any other factors here, like rest days or varying distances. The problem seems straightforward. So I think that's the correct answer.

Final Answer
James runs a total of $\boxed{540}$ meters a week. `<think>`
To determine the total distance James runs in a week, we can break the problem into smaller, logical steps.

Step 1: Determine distance per session
James runs 3 sprints in each session, and each sprint is 60 meters.
So, the distance he covers in one session is:
$3 \text{ sprints} \times 60 \text{ meters/sprint} = 180 \text{ meters}$
Step 2: Determine total sessions per week
James run 3 times a week, so we multiply the distance per session by the number of sessions per week:
$180 \text{ meters/session} \times 3 \text{ sessions/week} = 540 \text{ meters/week}$
Final Answer
James runs a total of 540 meters in a week. | `<think>` Okay, let's try to figure out how many meters James runs each week. He does three sprints three times a week. Each sprint is 60 meters. So, breaking it down, each time he does three sprints, that's three separate 60-meter runs. So, three sprints at 60 meters each would be 180 meters for each session. Then, since he does this three times a week, each week would add up to three sessions of 180 meters. So, three times 180 is 540 meters a week. Let me check again. Three sprints each time, three times a week. Each sprint is 60 meters. So per session, three sprints equals 180 meters. Three times a week, that's three sessions. So 180 meters three times is 540 meters total. That seems to add up correctly. I don't see any problems with that calculation. So the total is 540 meters per week. `<think>`
James runs 3 sprints 3 times a week, with each sprint covering 60 meters.
To calculate the total distance he runs each week:
Each session consists of 3 sprints, totaling 180 meters ($3 \times 60$ meters).
With three sessions per week, the total distance is:
3 sessions $\times$ 180 meters per session equals 540 meters.
Therefore, James runs a total of 540 meters per week. |

| | Qwen3-32B (Origin Model) | Qwen3-32B (Compressed by our LC-SVD) |
|---|---|---|
| **Question** | Every day, Wendi feeds each of her chickens three cups of mixed chicken feed, containing seeds, mealworms and vegetables to help keep them healthy. She gives the chickens their feed in three separate meals. In the morning, she gives her flock of chickens 15 cups of feed. In the afternoon, she gives her chickens another 25 cups of feed. How many cups of feed does she need to give her chickens in the final meal of the day if the size of Wendi's flock is 20 chickens? | |
| **Answer** | `<think>`
Okay, let's see. So Wendi has chickens, and she feeds them three times a day. Each chicken needs three cups of mixed feed per day. The problem is asking how many cups she needs to give in the final meal. She already gave them 15 cups in the morning and 25 cups in the afternoon. The flock has 20 chickens. Hmm.
First, maybe I should figure out the total amount of feed needed for all the chickens each day. Since each chicken gets three cups a day, and there are 20 chickens, that would be 3 cups per chicken times 20 chickens. Let me calculate that: 3 * 20 = 60 cups total per day.
So the total feed required for the day is 60 cups. Now, she's already given them some feed in the morning and afternoon. The morning was 15 cups and the afternoon was 25 cups. So combined, that's 15 + 25. Let me add that: 15 + 25 = 40 cups.
Therefore, she's already given 40 cups in the first two meals. Since the total needed is 60 cups, the remaining amount for the final meal would be 60 - 40. Let me subtract that: 60 - 40 = 20 cups.
Wait, so does that mean she needs to give 20 cups in the final meal? Let me check if I did everything right. Each chicken needs 3 cups a day. 20 chickens * 3 cups = 60 total cups. Morning and afternoon add up to 15 + 25 = 40. 60 - 40 is indeed 20. So yes, 20 cups in the final meal.
··· `<think>`
To determine how many cups of feed Wendi needs to give her chickens in the final meal of the day, we start by calculating the total amount of feed required for the entire day.
1. Total feed required per day:
···
Thus, Wendi needs to give her chickens 20 cups of feed in the final meal of the day. | `<think>` Okay, let me try to figure out this problem. Wendi has 20 chickens and feeds them three cups of feed each day. Each day, she gives them three meals. In the morning, she gives them 15 cups, and in the afternoon, another 25 cups. The question is how much feed she needs to give in the final meal of the day.
First, I need to figure out how much feed each chicken gets each day. The problem says that each chicken gets three cups of feed per day. Wendi has 20 chickens, so total feed per day is 20 times three cups. That would be 60 cups of feed per day. Then, the morning meal is 15 cups, and the afternoon meal is 25 cups. So, adding those together, that's 40 cups. The total feed per day is 60 cups, so the final meal would need to be 20 cups to reach the total of 60 cups.
Let me check my math again. 20 chickens times three cups per day equals 60 cups. Morning is 15 cups, afternoon is 25 cups, totaling 40 cups. The remaining amount needed is 20 cups for the final meal.
I think that's correct. So Wendi needs to give 20 cups of feed in the final meal of the day.
`<think>`
Wendi needs to give 20 cups of feed in the final meal of the day.
To summarize the solution:
Wendi's 20 chickens receive three cups of feed per day, totaling 60 cups. The morning and afternoon meals provide 15 and 25 cups respectively, totaling 40 cups. The remaining amount needed to reach the daily total of 60 cups is 20 cups. Therefore, Wendi needs to give 20 cups of feed in the final meal of the day. |

| | Qwen3-32B (Origin Model) | Qwen3-32B (Compressed by our LC-SVD) |
|---|---|---|
| **Question** | Something that has a long and sharp blade is a? 
 A. cup B. fan C. chuck D. sword E. spatula | |
| **Answer** | `<think>` 
 Okay, let's see. The question is asking for something that has a long and sharp blade. The options are A. cup, B. fan, C. chuck, D. sword, E. spatula. 
 First, I need to think about each option. A cup is a container for liquids, usually with a handle, but it doesn't have a blade. A fan is for moving air, maybe with blades, but those are usually flat and not sharp. Chuck is a term I'm not sure about in this context. Maybe a chuck is part of a machine, like a lathe chuck, which holds things in place, but not a blade. A sword is a weapon with a long, sharp blade. That fits. Spatula is a kitchen tool, usually with a flat blade, but it's not sharp, more for flipping food. So the answer should be D. Sword. 
 `<think>` 
 The correct answer is D. sword. 
 A sword is a weapon characterized by a long, sharp blade designed for cutting or thrusting. The other options do not fit the description: 
 cup is a container with no blade. 
 fan may have blades, but they are not sharp. 
 chuck is a mechanical device for holding objects, not a blade. 
 spatula has a flat, broad blade but is not sharp. 
 Thus, sword is the only item with a long, sharp blade. | `<think>` Okay, so I need to figure out which of these words refers to something that has a long and sharp blade. The options are cup, fan, chuck, sword, and spatula. Let me think about each one. A cup is a container for holding liquid, usually with a rounded top and no sharp edges. A fan is for moving air around, typically with blades that are not sharp. A chuck is a part of a machine or tool, maybe something that holds something, but I'm not sure what it looks like. A sword is a weapon with a long blade, usually sharp. A spatula is a kitchen tool for moving food around, typically with a broad, flat blade. 
 Based on that, the sword is the obvious choice because of its long and sharp blade. The other options don't fit the description. So I think the answer is D, sword. 
 `<think>` 
 The correct answer is D. sword. A sword is a weapon characterized by a long, sharp blade. The other options do not fit the description of having a long and sharp blade. 
 A. cup is a container for liquids and has no blade. B. fan has blades that are not sharp. C. chuck refers to a mechanical part and does not have a blade. E. spatula is a kitchen utensil with a broad, flat blade. 
 Therefore, D. sword is the correct answer. |

# D    MORE DETAILS ON STABLE OPTIMIZATION IN LAYER COLLABORATIVE WEIGHT DECOMPOSITION

In our proposed *Layer Collaborative Weight Decomposition* framework, we introduce a learnable whitening matrix $\mathbf{S}$ for each matrix to capture the coordinated, nonlinear interactions among weight matrices within a Transformer block. The optimization objective is formulated at the functional block level:

$$\min_{\mathcal{S}} \frac{1}{N} \sum_{i=1}^{N} \|f(\mathbf{X}_i; \mathcal{W}) - f(\mathbf{X}_i; \hat{\mathcal{W}})\|_F^2, \tag{VI}$$

$$\hat{\mathbf{W}}_k = \text{SVD}(\mathbf{W}_k \mathbf{S}_k; r) = \mathbf{U}_{:,:r} \mathbf{\Sigma}_{:r,:r} \mathbf{V}_{:r,:}^\top \mathbf{S}_k^{-1}. \tag{VII}$$

where $f(\cdot; \mathcal{W})$ denotes a transformer sub-block (e.g., MLP or MHA), $\mathcal{W} = \{\mathbf{W}_k\}_{k=1}^K$ are the original weights, and $\hat{\mathcal{W}} = \{\hat{\mathbf{W}}_k\}_{k=1}^K$ are their low-rank approximations defined via the data-aware SVD in Eq. (VII). This formulation enables end-to-end optimization of the whitening matrices

$\mathcal{S} = \{\mathbf{S}_k\}_{k=1}^K$ through gradient descent, allowing the decomposition to adapt to both the input data distribution and the compositional structure of the block.

However, since the approximated weights $\hat{\mathbf{W}}_k$ are computed via truncated SVD, the backward pass requires differentiation through the SVD operation, which introduces potential numerical instabilities in gradient computation. Specifically, the gradient of a matrix function involving SVD depends on a critical matrix $\mathbf{K}$ whose entries are defined as:

$$\mathbf{K}_{ij} = \begin{cases} \dfrac{1}{\sigma_j^2 - \sigma_i^2}, & i \neq j, \\ 1, & i = j, \end{cases} \tag{VIII}$$

where $\sigma_i$ and $\sigma_j$ denote the $i$-th and $j$-th singular values of the matrix $\mathbf{W}_k\mathbf{S}_k$. When two singular values $\sigma_i$ and $\sigma_j$ become very close, particularly common in LLM high-dimensional weight matrices (Wang et al., 2019; Marčenko & Pastur, 1967), the denominator $\sigma_j^2 - \sigma_i^2$ approaches zero, causing $\mathbf{K}_{ij}$ to grow arbitrarily large or become undefined. This phenomenon can lead to exploding gradients, hampering convergence and resulting in suboptimal solutions.

To mitigate this issue, we adopt a robust gradient approximation strategy (Wang et al., 2021). We handle the problematic term $1/(\sigma_j^2 - \sigma_i^2)$ in two cases based on the magnitude of the singular values:

- **Case 1: $\sigma_i \approx \sigma_j > 0$.** In this case, we rewrite the term as:

$$\frac{1}{\sigma_j^2 - \sigma_i^2} = \frac{1}{\sigma_j^2(1-r)}, \quad \text{where } r = \frac{\sigma_i^2}{\sigma_j^2}.$$

  We then approximate $1/(1-r)$ using the $C$-th order Taylor expansion $\frac{1}{1-r} \approx \sum_{c=0}^C r^c$, which yields:

$$\frac{1}{\sigma_j^2 - \sigma_i^2} \approx \frac{1}{\sigma_j^2} \sum_{c=0}^C \left( \frac{\sigma_i^2}{\sigma_j^2} \right)^c \approx \frac{C+1}{\sigma_j^2}.$$

  This bounded approximation prevents uncontrolled growth when $\sigma_i \to \sigma_j$, and the truncation order $C$ serves as a hyperparameter controlling the trade-off between accuracy and stability.

- **Case 2: $\sigma_i \approx \sigma_j \approx 0$.** When both singular values are close to zero, even the Taylor expansion may fail due to division by small $\sigma_j^2$. In this regime, we replace the denominator with a small positive constant $\lambda > 0$:

$$\frac{1}{\sigma_j^2 - \sigma_i^2} \approx \frac{1}{\lambda}.$$

  This clipping ensures numerical tractability without significantly affecting the low-rank approximation, as components associated with near-zero singular values contribute negligibly to the reconstructed matrix.

Using this stabilized gradient computation, the full gradient with respect to the matrix $\mathbf{A} = \mathbf{W}_k\mathbf{S}_k$ can be expressed as:

$$\mathbf{g}_A = \mathbf{U} \left[ \frac{\text{skew}(\mathbf{U}^\top \mathbf{g}_U)}{\mathbf{K}} \circ \boldsymbol{\Sigma} + \boldsymbol{\Sigma} \circ \frac{\text{skew}(\mathbf{V}^\top \mathbf{g}_V)}{\mathbf{K}} + \text{diag}(\mathbf{g}_{\boldsymbol{\Sigma}}) \right] \mathbf{V}^\top, \tag{IX}$$

$$\text{where} \quad \mathbf{K}_{ij} = \begin{cases} \dfrac{1}{\sigma_j^2} \sum_{c=0}^C \left( \dfrac{\sigma_i^2}{\sigma_j^2} \right)^c, & \text{if } \sigma_i \approx \sigma_j > 0, \\ \dfrac{1}{\lambda}, & \text{if } \sigma_i \approx \sigma_j \approx 0, \\ \dfrac{1}{\sigma_j^2 - \sigma_i^2}, & \text{otherwise}, \end{cases} \tag{X}$$

where $\mathbf{K}_{ii} = 1$ for all $i$; $\mathbf{g}_U$, $\mathbf{g}_V$, and $\mathbf{g}_{\boldsymbol{\Sigma}}$ are the gradients of the loss with respect to the left singular vectors, right singular vectors, and singular values, respectively; $\text{skew}(\cdot)$ denotes the skew-symmetric part of a matrix; $\circ$ denotes element-wise multiplication (Hadamard product); and the division by $\mathbf{K}$ is performed element-wise using the stabilized definition above.

