# OpenReview forum: "Layer Collaborative Low-Rank Decomposition with Automatic Rank Search for LLM Compression"
_ICLR.cc/2026/Conference — Submitted to ICLR 2026_

### Official Review · Reviewer_fa67 · 2025-10-30

**Soundness:** 3
**Presentation:** 3
**Contribution:** 2
**Rating:** 6
**Confidence:** 3

**Summary:**

The paper proposes a post-training compression method called LC-SVD. The method augments standard per-layer SVD with a learnable whitening matrix so that the decomposition better matches the actual calibration data and reduces block-level error accumulation. It further introduces an error-driven rank search that first estimates block sensitivities from perplexity under light compression, then samples non-uniform rank configurations accordingly, and finally selects the best one using calibration perplexity. The paper also details several optimization choices, such as an adaptive stopping criterion and a numerically stable procedure for learning the whitening matrices. Empirical results and ablations show that each component contributes to the final performance.

**Strengths:**

* The approach is described in sufficient detail, and the paper is clearly written, with experimental settings and evaluation protocols laid out in an organized way.
* The detailed ablation study showcases the contribution from each component of the method.
* The topic of post-training compression is practically valuable and interesting, since it could avoid the catastrophic-forgetting issues that can arise in fine-tuning–based compression methods.

**Weaknesses:**

* Structure-aware (or “layer-collaborative,” as the paper calls it) compression/fine-tuning is not new in this area. Prior work such as [1] and [2] has already explored adaptive, layer-dependent low-rank allocation. The related-work section would be stronger with a more complete positioning against these lines of work.
* Evaluating mainly at 80% parameter retention is not very compelling, since many recent pruning/quantization/compression methods operate at much lower budgets (although typically they are task-oriented, not general-purpose models as the paper investigates). The appendix shows that performance drops substantially when the budget is pushed down, which raises a natural question: in the low-budget regime, is this post-training SVD approach actually preferable to modern quantization or task-specific fine-tuning/pruning pipelines? I would like to hear the authors' thoughts on this.
* The paper notes that previous methods “lack theoretical optimality under rank constraints,” but the proposed method itself does not provide much theoretical analysis either.

[1] Zhang et al; AdaLoRA: Adaptive Budget Allocation for Parameter-Efficient Fine-Tuning.
[2] Hua et al; Dynamic Low-rank Estimation for Transformer-based Language Models.

**Questions:**

Does the choice of calibration data affect the final results? For instance, how sensitive is the method to the size of the calibration set or to shifting its source domain?

---

### Official Review · Reviewer_Ee2f · 2025-10-30

**Soundness:** 2
**Presentation:** 3
**Contribution:** 2
**Rating:** 2
**Confidence:** 5

**Summary:**

The paper proposes LC-SVD, a layer collaborative low-rank decomposition framework with automatic rank search for efficient compression of Large Language Models (LLMs). Unlike conventional SVD-based methods that compress each layer independently, LC-SVD jointly compresses all linear layers within a transformer block to preserve cross-layer dependencies and reduce error accumulation. It further introduces an error-driven rank search strategy that allocates ranks adaptively based on each block’s sensitivity to compression, avoiding suboptimal heuristic rank distributions.

**Strengths:**

1. This work focuses on an important topic — compressing large models for deployment in resource-constrained environments.
2. The motivations behind the two proposed improvements are clear: one aims to capture cross-layer interactions during compression, while the other addresses the optimal allocation of parameter budgets in low-rank decomposition.

**Weaknesses:**

1. For “Layer Collaborative Weight Decomposition”, the idea of using gradient descent to learn a whitening matrix seems questionable. If gradient-based optimization is feasible, then directly fine-tuning the pruned low-rank weights would be more straightforward and effective. Even if global backpropagation is impractical, one could still fine-tune each block separately or employ gradient checkpointing to reduce memory overhead. Moreover, the proposed approach appears computationally expensive, since it requires performing SVD at every iteration, as the matrix $S$ changes in each step. In contrast, all compared SVD-based methods in the experiments adopt one-shot pruning without gradient descent. For a fair comparison, these baselines should also include fine-tuning to match the proposed method’s training cost and optimization benefits.

2. For “Error-Driven Rank Search”, the idea of estimating each block’s sensitivity to determine non-uniform sparsity is not novel. Similar concepts have already been explored in prior works such as OWL [1] and ALS [2] for unstructured pruning, as well as ASVD [3] and WeLore [4] for SVD-based compression. To convincingly demonstrate the effectiveness of the proposed rank search, the paper should compare against these non-uniform sparsity allocation baselines rather than only using a uniform sparsity baseline (at least some of them). Moreover, the proposed approach appears also heuristic, by sampling candidate configurations under a prior and selecting the best-performing one. In addition, the improvement brought by this module is quite limited: as shown in the results, the perplexity only decreases slightly from 7.09 (with Layer Collaborative SVD) to 6.95 (full LC-SVD).

3. More recent state-of-the-art SVD-based compression methods, such as Pivoting Factorization [5], are not included in the comparison, which limits the completeness of the experimental evaluation.

4. Typo: Only 1) is written in abstract.


[1] Yin, Lu, et al. "Outlier weighed layerwise sparsity (OWL) a missing secret sauce for pruning LLMs to high sparsity." Proceedings of the 41st International Conference on Machine Learning. 2024.

[2] Li, Wei, et al. "Adaptive layer sparsity for large language models via activation correlation assessment." Advances in Neural Information Processing Systems 37 (2024): 109350-109380.

[3] Yuan, Zhihang, et al. "Asvd: Activation-aware singular value decomposition for compressing large language models." arXiv preprint arXiv:2312.05821 (2023).

[4] JAISWAL, AJAY KUMAR, et al. "From Low Rank Gradient Subspace Stabilization to Low-Rank Weights: Observations, Theories, and Applications." Forty-second International Conference on Machine Learning.

[5] Zhao, Jialin, Yingtao Zhang, and Carlo Vittorio Cannistraci. "Pivoting Factorization: A Compact Meta Low-Rank Representation of Sparsity for Efficient Inference in Large Language Models." Forty-second International Conference on Machine Learning.

**Questions:**

Check above.

---

### Official Review · Reviewer_zuFu · 2025-10-30

**Soundness:** 3
**Presentation:** 3
**Contribution:** 2
**Rating:** 4
**Confidence:** 4

**Summary:**

This paper proposes LC-SVD for compressing LLMs via low-rank decomposition. The method consists of two components: (1) block-wise collaborative decomposition that optimizes all weight matrices within a transformer block jointly, and (2) error-driven rank search that adaptively allocates ranks based on block sensitivity.  However, the "collaborative decomposition" is actually standard joint optimization where each matrix still receives its own independent SVD, with only the whitening matrices being updated via a shared block-level loss. The "error-driven rank search" is an expensive brute-force procedures, taking one hour on GPU.

**Strengths:**

Solid experimental setting: Comprehensive evaluation across 3 models, 3 compression ratios, 2 language modeling benchmarks, and 6 downstream tasks, with thorough ablation studies and reproducible implementation details.

**Weaknesses:**

1. Overstated Novelty - "Collaborative Decomposition" is Standard Joint Optimization.  All whitening matrices are updated based on the same block reconstruction loss. Each matrix still gets its own independent SVD; the only "joint" aspect is that optimization uses block-level loss instead of layer-level loss in previous work. This is a few-lines code change—using block output instead of layer outputs for the loss function.
2. "Error-Driven Rank Search" seems to be a brute-force hyperparameter search. Prior work already does non-uniform rank allocation; this just uses a different (expensive) search procedure. For example, previous work RankDyna (EMNLP 2023 Findings https://aclanthology.org/2023.findings-emnlp.621/), demonstrates adaptive allocation during training surpasses fixed post-hoc assignment.

**Questions:**

1. How does static post-hoc search compare to RankDyna's dynamic allocation? maybe adding something like: "Unlike methods that compress during task-specific fine-tuning (e.g., RankDyna), our approach targets post-hoc compression for general deployment without additional training. This is valuable when: (1) task data is unavailable, (2) training compute is constrained, or (3) a general-purpose compressed model is needed for multiple downstream applications without per-task adaptation."

2. What is the total compression time (optimization + 1 hour rank search) vs. baselines?

---

### Official Review · Reviewer_mr2G · 2025-10-31

**Soundness:** 2
**Presentation:** 3
**Contribution:** 1
**Rating:** 2
**Confidence:** 5

**Summary:**

To mitigate reconstruction error brought by model decomposition, this paper propose propose LC-SVD, a layer collaborative SVD framework with automatic rank search that enables adaptive low-rank compression of LLMs. LC-SVD mainly includes two parts: updating whitening matrix and rank allocation. Extensive and comprehensive experiments are carried out to demonstrate the superiority of the proposed method.

**Strengths:**

1. This paper is well written and easy to follow.

2. The experiments carried out in this paper are extensive and comprehensive. But it is also limited, details refer to Weaknesses below.

**Weaknesses:**

1. Limited evaluation and weak baselines. SVD-LLM is outdated, and should be replaced its successor SVD-LLM V2. The evaluation lacks the comparison with other non-decomposition-based strong baseline. Additionally, experiments should also include other non-MoE models from different LLM family. As for the baseline, authors are suggested to browse the publication in recent conferences.

2. Lack of novelty. The contribution of this paper is just the combination of updating whitening matrix and rank allocation, and there are so many similar works in this field. The rank search part is heuristic-based and prune to be unsound. It needs more analysis and evaluation to demonstrate its generalizability.

3. Incremental contributions. At least as for myself, this paper doesn't contribute any new insight on this area.

**Questions:**

See Weaknesses.

---

### Meta-Review · Area_Chair_jfce · 2026-01-12

**Summary:**

The overall evaluation leans toward rejection, with two reviewers recommending rejection, one weak reject, and one weak accept. Reviewers generally agree that the paper is clearly written and supported by extensive experimental results and ablations, demonstrating careful empirical work. However, the main weaknesses consistently identified are limited novelty, incremental contributions, and incomplete or outdated baselines, particularly missing comparisons to more recent SVD-based and non-decomposition-based compression methods. Several reviewers also question the soundness and generalizability of the heuristic, computationally expensive error-driven rank search, and whether the proposed “layer collaborative” formulation meaningfully differs from standard block-level joint optimization.

**Reviewer Concerns:**

No rebuttal was provided, and thus none of the reviewers’ concerns were addressed.

**Reviewer Scores:**

No rebuttal was provided, and thus none of the reviewers’ concerns were addressed.

Reviewer mr2G: 2
Reviewer zuFu: 4
Reviewer Ee2f: 2
Reviewer fa67: 6

---

### Decision · Program_Chairs · 2026-01-26

Reject